# Nutritional Quality of Milk Fat from Cows Fed Full-Fat Corn Germ in Diets Containing Cactus Opuntia and Sugarcane Bagasse as Forage Sources

**DOI:** 10.3390/ani13040568

**Published:** 2023-02-06

**Authors:** Camila S. da Silva, Marco A. S. Gama, Erick A. M. Silva, Emília F. Ribeiro, Felipe G. Souza, Carolina C. F. Monteiro, Robert E. Mora-Luna, Júlio C. V. Oliveira, Djalma C. Santos, Marcelo de A. Ferreira

**Affiliations:** 1Department of Animal Science, Federal Rural University of Pernambuco, Dom Manoel de Medeiros Street, Dois Irmãos, Recife 52171-900, Brazil; 2Embrapa Southeast Livestock, Washington Luiz Road, Km 234, São Carlos 13560-970, Brazil; 3Department of Animal Science, State University of Alagoas, BR 316, Km 87.5, Santana do Ipanema 57500-000, Brazil; 4Department of Animal Science, Federal University of North of Tocantins, Araguaína 77804-970, Brazil; 5Agronomic Institute of Pernambuco, 100 Edílio Simões da Rocha St, Arcoverde 56500-000, Brazil

**Keywords:** co-products, crossbred cows, fat supplementation, milk fat, *Opuntia* sp., semi-arid

## Abstract

**Simple Summary:**

Lipid supplementation is a common practice in dairy cow nutrition to control heat stress, provide more energy, and alter the fatty acid profile of milk fat. Full-fat corn germ (FFCG)—a by-product from corn processing—has been indicated as an alternative lipid ingredient because of its positive effects on milk production and milk fatty acid profile. Yet, FFCG may cause ruminal disturbances that lead to milk fat depression (MFD) in cows fed fresh sugarcane and cactus cladodes, which are important forage resources for the supplementation of dairy herds in the semi-arid region. Our most recent results show that replacing fresh sugarcane with sugarcane bagasse in the diet prevents the MFD observed in cows fed FFCG. The association of FFCG, cactus cladodes, and sugarcane bagasse improves milk production and milk fat content and alters the composition of milk fat towards a healthier fatty acid profile. Hence, FFCG supplementation may be advantageous to milk producers through improved cow performance while promoting greater fat yield and fat quality for the manufacturing of dairy products with potential health benefits.

**Abstract:**

We evaluated the performance, milk composition, and milk fatty acid profile of cows fed diets composed of cactus cladodes (*Opuntia stricta* [Haw.] Haw), sugarcane bagasse and increasing levels of full-fat corn germ (FFCG). We hypothesized that ground corn can be effectively replaced by FFCG when cactus cladodes and sugarcane bagasse are used as forage sources. The cows were randomly distributed into two 5 × 5 Latin Squares and fed five diets in which ground corn was progressively replaced with full-fat corn germ (FFCG; 0%, 25%, 50%, 75%, or 100% of substitution). Adding FFCG to the diet increased milk production and milk fat content and reduced milk protein content. Overall, FFCG reduced the proportion of saturated FAs and increased mono- and polyunsaturated FAs in milk, including CLA isomers. In addition, activity indices of stearoyl-CoA desaturase were reduced by increasing levels of FFCG. We conclude that the substitution of corn for FFCG in diets based on cactus cladodes and sugarcane bagasse positively modifies the FA profile of milk and could add commercial value to milk products (e.g., CLA-enriched milk). In addition, the milk fat response indicates that the basal diet was favorable to the rumen environment, preventing the *trans-10* shift commonly associated with milk fat depression.

## 1. Introduction

Dairy farming is one of the few viable activities in the Brazilian northeastern semi-arid region [1]. However, as the dairy sector has been affected by escalating marketing prices of corn [2], finding alternative low-cost energy sources has become even more critical. Meanwhile, the corn processing industry generates a co-product known as full-fat corn germ (FFCG), a solid, lipid-rich material with a significant concentration of unsaturated fatty acids (FA), mainly 18:2 n-6 and *cis*-9 C18:1 [3]. Moreover, FFCG presents a high oxidative stability and competitive price compared to other commonly marketed lipid supplements.

Lipid sources can be strategically fed to lactating cows raised in warm regions to replace carbohydrate sources, enabling manipulation of energy density in the diet while aiding in the control of heat stress [4,5]. Depending on the lipid composition of the supplement, these ingredients can also improve the FA profile of milk fat [6].

Previous results demonstrate that FFCG can be used to supplement high-producing dairy cows fed diets based on corn silage and alfalfa hay [4]. However, it has been acknowledged that roughage type plays a significant role in the productive responses of lipid-fed cows because the effects of lipid feeding are influenced by basal diet composition [7]. Furthermore, milk composition and milk FA profile are considerably affected by the interaction between different lipid supplements and roughages added to the diet [8,9,10].

Cactus cladodes (*Opuntia* sp.) are one of the most important forages fed to dairy herds in the semiarid region of Brazil [1]. Recent evidence indicates that cactus cladodes modulate the ruminal biohydrogenation (BH) of dietary polyunsaturated fatty acids (PUFA), favoring the incorporation of potentially health-beneficial FA (e.g., *trans*-11 C18:1 and *cis*-9, *trans*-11 CLA) into the milk fat of dairy cows [11]. This effect has been associated, at least in part, with appreciable amounts of phenolic compounds in cactus cladodes.

Although initial studies indicate the potential of FFCG as an alternative lipid ingredient for dairy cows [3,4], adding FFCG in diets containing fresh sugarcane and cactus cladodes alters the normal ruminal BH pathway and induces the milk fat depression syndrome [3]. Given this issue, the present study was designed to test the hypothesis that replacing corn with FFCG could improve milk production and promote positive changes in the FA profile of milk of cows fed diets composed of cactus cladodes (*Opuntia stricta* [Haw.] Haw) and sugarcane bagasse as forage sources.

## 2. Materials and Methods

The study was conducted according to ethical standards approved by the Ethics Committee on Animal Welfare of the Federal Rural University of Pernambuco (UFRPE), license No. 7149140220. The experiment took place in Arcoverde, state of Pernambuco, Brazil (latitude −8.35, longitude −36.84), from January 27 to 10 May 2020.

### 2.1. Animals and Facilities

Ten multiparous Girolando cows, with an initial body weight of 500 ± 66 kg and 90 ± 15 days in milk, were used in the experiment. All cows were adapted to management, facilities, and diet ingredients for 15 days before the onset of the trial (pre-experimental period). Subsequently, the cows were distributed into two 5 × 5 Latin Squares with 21-day experimental periods; within a period, the first 14 days were used for adaptation of the cows to experimental diets and the last 7 days for data and sample collection. Hence, the total experimental period lasted 105 days. The animals were allocated to each Latin square based on their daily milk production at the end of the pre-experimental period.

The cows were housed in individual stalls (24 m^2^) separated by galvanized steel wire. The stalls were equipped with individual feeders and shared drinking troughs. Each stall contained a concrete-floor area (where feeders were placed) and a sand-bedding area for resting.

### 2.2. Experimental Diets

Dietary treatments were represented by five levels of substitution (0%, 25%, 50%, 75%, and 100%) of ground corn for full-fat corn germ (FFCG; Table 1). Cactus cladodes and sugarcane bagasse were used as roughage sources in the diets and kept in equal proportions across dietary treatments (~42 and 32% of DM, respectively).

The proportion of roughages and concentrate in the experimental diets were adjusted to meet the nutritional requirements of cows with an average daily milk production of 15 kg/day and 3.5% milk fat [7]. Diets were fed *at libitum* as a total mixed ration (TMR) at 07:00 and 16:00 h. The amount of TMR provided was daily adjusted to allow 5–10% of feed refusals, which were weighed and recorded before the morning feeding.

### 2.3. Milk Sampling and Analysis

The volume of milk produced at two daily milkings (06:30 and 15:30 h) was recorded throughout the data collection period (days 15 to 21). Individual milk samples were collected on days 19 and 20 of each experimental period, corresponding to 1% of the milk volume produced at each milking (two samples per day). The samples were kept under refrigeration until the last milk sampling on day 20 and combined per animal and period thereafter. A 50-mL aliquot of the composite sample was added to a Falcon tube containing preservative (Bronopol^®^) and kept at 2–6 °C until analyzed for fat, protein, and total solids within 5 days. Another set of composite milk samples (5 mL, preservative-free) was submitted for analysis of the fatty acid (FA) profile of milk.

Milk samples were thawed at room temperature and a 1-mL aliquot was used for lipid extraction [13]. The extracted lipids were dissolved in hexane and methyl acetate and further transesterified using a solution of sodium methoxide in methanol [14]. The mixture was neutralized with oxalic acid (1 g of the acid in 30 mL of diethyl ether), and methanol residues were removed by adding calcium chloride. The fatty acid methyl esters (FAME) were then separated and quantified using a 7820A gas chromatograph (Agilent Technologies, Santa Clara, CA, USA) equipped with FID and a CP-Sil 88 fused silica capillary column (100 m × 0.25 mm × 0.2 μm; Varian, Mississauga, ON, Canada). The equipment was operated according to Reference [15].

The FAME contained in milk samples was identified by comparison with the retention time of commercial standards (Supelco**^®^** 37 Component FAME Mix, Sigma-Aldrich, Bellefonte, USA; Larodan Ab, Stockholm, Sweden; Luta-Cla**^®^** 60, BASF AG, São Paulo, Brazil). *Trans/cis* C18:1 isomers and *trans*-9, *cis*-11 CLA were identified according to their reported elution orders under the same analytical conditions described elsewhere [15]. The FA composition of milk was expressed as a proportion of total FAs using theoretical response factors [16]. The indices of activity for the enzyme stearoyl-CoA desaturase-1 (SCD-1) were estimated as the ratio of the enzyme’s products and substrates [17].

### 2.4. Statistical Analysis

One of the cows presented health issues during the last experimental period. Hence, data corresponding to this cow within the fifth period was estimated as lost observations. The experimental data were submitted for analysis of variance and regression using the MIXED procedure of SAS (version 9.4) according to the following statistical model:Yijkl = μ + Ti + Qj + Pk + (A/Q)lj + εijkl
where: Yijkl = observation ijkl; μ = overall mean; Ti = fixed effect of treatment i; Qj = fixed effect of square j; Pk = fixed effect of period k; (A/Q)lj = random effect of animal l within square j; εijkl = random error with mean 0 and variance σ2.

Means obtained for the different levels of corn substitution (G0, G25, G50, G75, and G100) were compared from the unfolding of the sum of squares into orthogonal contrasts and adjustments of the regression equations. Linear and quadratic effects were considered statistically significant when *p* < 0.05.

## 3. Results

Substitution of corn for full-fat corn germ (FFCG) had a quadratic effect on milk production (*p* = 0.0183; Table 2).

The addition of up to 75% of corn germ in the diet (G0 to G75) increased milk production from 13.37 kg/day to 14.93 kg/day (+1.5 kg), whereas complete substitution (G100) declined milk production by 0.85 kg/day. A linear increase (*p* = 0.0387) in milk fat content was observed in response to FFCG feeding. Milk protein content changed in a quadratic manner (*p* = 0.0106), decreasing by 0.18 percentage units when corn was replaced by up to 75% FFCG (G75), followed by a slight increase (0.09 percentage units) at the highest FFCG level (G100). Dietary treatments did not influence the content of total solids in milk (*p* > 0.05).

Replacing corn with FFCG significantly modified the proportions of several fatty acids in milk fat (Table 3). Except for C4:0, C5:0, and *trans*-9 C16:1, there was a linear (*p* < 0.05) decline in proportions of short-chain (C4:0 to C10:0) and medium-chain (C12:0 to C16:0) fatty acids in milk. Similarly, the proportions of ≥C20 fatty acids (except C20:0 and C20:1 n-9) were linearly reduced (*p* < 0.05) by the inclusion of FFCG in the diet.

Among C18 fatty acids secreted in the milk (Table 4), only α-linolenic acid (C18:3 n-3) was linearly reduced by treatments (*p* < 0.05). The proportions of stearic (C18:0), oleic (*cis*-9 C18:1), and *cis*-12 C18:1 acid responded quadratically to FFCG feeding (*p* < 0.05). Other C18 fatty acids, such as linoleic (*cis*-9, *cis*-12 18:2), vaccenic (*trans*-11 18:1), and rumenic (*cis*-9, *trans*-11 CLA) increased linearly (*p* < 0.05) or were unaffected by treatments (C18:3 n-6 and *cis*-11 C18:1; *p* > 0.05).

The effects of FFCG addition on the concentration of C18 acids contributed to a linear or quadratic (*p* < 0.05) increase in the proportion of monounsaturated (MUFA) and polyunsaturated fatty acids (PUFA) in milk (Table 5). Conversely, FFCG intake linearly (*p* < 0.05) reduced the proportions of omega-3 fatty acids (n-3), short-chain fatty acids (SCFA), medium-chain fatty acids (MCFA), saturated fatty acids (SFA), odd-chain fatty acids (OCFA), branched-chain fatty acids (BCFA), and odd- and branched-chain fatty acids (OBCFA) in milk fat. The proportion of omega-6 (n-6) fatty acids was not altered by FFCG (*p* > 0.05).

The fatty acid ratios (*trans*-C18:1/C18:0, *trans*-11 C18:1/C18:0, and n-6/n-3) increased linearly (*p* < 0.0001) as FFCG replaced corn (Table 5). Furthermore, there was a linear decline (*p* < 0.05) in three of the four SCD-1 activity indices evaluated (SCD_14_, SCD_16_, and SCD_CLA_), while SCD_18_ (*cis*-9 C18:1/C18:0 pair) showed a quadratic effect of treatments (*p* = 0.0254).

## 4. Discussion

The effects of lipid supplementation on dairy cows’ performance and milk composition are variable. Nonetheless, a meta-analysis performed to address the effects of supplementation of different lipid sources on dairy cows [18] identified a pattern of increased milk production in lipid-supplemented animals. This observation is consistent with our initial hypothesis (i.e., replacing corn with FFCG could raise milk production).

The quadratic increase in milk production in our study was likely a response to the increased energy density of the diet caused by the addition of FFCG, as the inclusion of fat sources often enhances milk production because of the additional energy supply [7]. Additionally, fat supplementation affects milk production in a curvilinear manner [19], as we observed in the present study. A similar trend was found for dairy cows supplemented with FFCG at up to 21% [20] and 16% of diet DM [3].

The composition of experimental diets (Table 1) shows that the EE content of the diets gradually increased with the inclusion of FFCG, reaching 7.4% DM. This value approximates the limit of EE intake for ruminants (7.0% DM) [21]. It is possible that the lipid intake in treatments G75 and G100 hindered the activity of rumen microorganisms and digestion of the fibrous fraction to some extent, consequently depressing feed intake and affecting milk production. In fact, reduced DM intake has been observed when ground corn was fully replaced with FFCG in a diet based on cactus cladodes and fresh sugarcane [3].

One of the concerns around lipid supplementation to dairy cows is the milk fat depression syndrome (MFD), in which milk production and fat content decline without apparent changes in milk yield [22]. However, the significant linear effect of treatments on milk fat content (Table 2) shows that increasing levels of FFCG benefits milk fat secretion under similar dietary conditions.

A study conducted by [23] demonstrated that the dietary concentration of fiber and fermentable carbohydrates is a determinant factor to offset the negative impact of unsaturated lipids on rumen function and preventing MFD. For instance, there was a sharp drop in milk fat concentration (from 3.49% to 2.23%) when diets based on fresh sugarcane, cactus cladodes, and FFCG were offered to Holstein cows [3]; in the present study (Girolando cows), using sugarcane bagasse as a source of NDF increased milk fat concentration.

It is not clear whether this difference between studies was caused by adverse associative effects of fresh sugarcane/bagasse, cactus cladodes, and FFCG in the rumen; by the amount of FFCG supplemented (slightly lower in the current experiment); by dietary apNDF concentration (10.9% higher in the present study); or a possible sensitivity of Holstein cows to MFD compared to crossbred cows. However, previous observations [23] and the percentage values of milk fat found herein point out that the amount and effectiveness of fiber present in sugarcane bagasse may have been key to maintaining an appropriate concentration of fat in milk. Moreover, one of the outstanding characteristics of forage cactus is its high pectin content [24], which favors acetic fermentation and supply of acetate and butyrate for fat synthesis in the mammary gland [25].

Regarding the effect of FFCG intake on milk protein concentration, we can assume that the quadratic response in protein concentration is merely a dilution effect because milk yield also showed a quadratic behavior. This is often observed in dairy cows supplemented with lipid sources [4,26,27,28,29], as milk protein concentration reflects milk volume and milk protein production [30]. Moreover, the substitution of non-fiber carbohydrate sources for lipid-rich ingredients can result in glucose deficiency in rumen bacteria, compromising microbial protein synthesis and amino acid availability for milk protein synthesis [31]. The depressing effect of lipid supplementation on DM intake also diminishes the supply of amino acids for milk protein synthesis in the mammary gland [28].

Changes in the proportions of the different fatty acids present in milk (Table 3) show that the unsaturated fatty acids present in FFCG (e.g., C18:2 n-6 and, to a lesser extent, *cis*-9 C18:1) were partially available for ruminal BH, favoring the secretion of desirable fatty acids in milk such as *trans*-11 C18:1 and *cis*-9, *trans*-11 CLA.

The decrease in <C16 milk fatty acids (Table 3) corroborates with the <C16 fatty acid content (except the sum of *trans* C16:1) in cows supplemented with sunflower oil (50 g/kg of diet) and fed low concentrate levels (35% of total DM) [32]. It has been estimated that approximately 90% of the myristoleic acid (*cis*-9 C14:1) and between 50 and 56% of palmitoleic acid (*cis*-9 C16:1) present in bovine milk originates from endogenous synthesis by the mammary stearoyl-CoA desaturase (SCD-1) [33]. Thus, the linear decrease in <C16 fatty acids indicates inhibition of endogenous synthesis of short and medium-chain fatty acids due to a higher dietary intake of long-chain fatty acids (especially C18:2 n-6) in FFCG treatments, which is a typical response [34].

Odd-chain and branched-chain fatty acids (OBCFA), such as *iso* C17:0 and *anteiso* C17:0, derive mainly from the lipid membrane of rumen bacteria [35,36]. As a result, bacterial growth rate and bacterial concentration of *iso* C17:0 are strongly correlated [36]. In addition, ruminal infusion of soybean oil reduced the proportion of OBCFA in milk [37], suggesting that lipid supplementation stimulates the uptake of dietary fatty acids by rumen bacteria.

Hence, the individual values and the sum of odd-chain and branched-chain fatty acids shown in Table 3 and Table 5 suggest that: (i) replacing corn with FFCG, at levels tested in our study, suppress the growth of the rumen microbiota but does not completely inhibit the enzymatic activity of bacteria capable of biohydrogenation; and/or inhibits synthesis of new bacterial fatty acids by stimulating incorporation of dietary lipids [36]. Furthermore, the reduced proportion of OBCFA by FFCG diets may be partially linked to a possible effect of FFCG on protozoa, which are susceptible to the antimicrobial effects of unsaturated lipids [38] and possess higher concentrations of certain branched-chain fatty acids compared to bacteria [39].

The proportions of C18:0 (final product of rumen biohydrogenation) and *cis*-9 C18:1 (coming mostly from endogenous mammary synthesis via the action of SCD-1 on C18:0) in the milk of lipid-fed cows depend significantly on the proportion of long-chain fatty acids contained in the supplement [40] and composition of the basal diet [11]. Given that FFCG contains predominantly long-chain polyunsaturated fatty acids [3], one would expect that its progressive addition to the diet would linearly increase the formation of C18:0 and, consequently, the proportions of C18:0 and *cis*-9 C18:1 in milk fat due to substrate availability for rumen biohydrogenation and increased C18:0 supply to the mammary gland. Instead, we observed a quadratic increase in the proportions of C18:0 and *cis*-9 C18:1 in milk fat (Table 4), as did [3]. The explanation for this behavior seems to lie in the utilization of cactus cladodes for diet formulation in both studies.

The link between cactus cladodes and reduced milk C18:0 concentration was first suggested in a study where partial replacement of sorghum silage with cactus cladodes (*Opuntia stricta*) for cows supplemented with soybean oil (2.7% DM) significantly reduced the proportions of *cis*-9 C18:1 and C18:0 in milk fat [11]. The authors suggested that phenolic compounds present in the cladodes [41] modulate rumen biohydrogenation through manipulation of the composition of the rumen microbiota; as a result, greater amounts of *trans*-11 C18:1 leave the rumen at the expense of C18:0 [42]. This effect is consistent with the linear increase in the *trans*-11 C18:1/C18:0 ratio in milk fat of cows in the present study (Table 5).

Endogenous mammary synthesis by the enzyme SCD-1, using *trans*-11 C18:1 from ruminal BH as substrate, is responsible for most of the *cis*-9, *trans*-11 CLA secreted in milk [43]. Therefore, the linear decline in the activity index of SCD-1 for the *cis*-9, *trans*-11 CLA/*trans*-11 C18:1 pair (SCD_CLA_) observed in response to the addition of corn germ in the diets (Table 5) suggests a reduction in endogenous *cis*-9, *trans*-11 CLA synthesis. As observed for SCD_CLA_, there was also a linear drop in the SCD_14_ index, considered the best indicator of SCD-1 activity in the mammary gland [44]. Hence, the substitution of corn for FFCG may have, in fact, reduced the activity of this enzyme in the mammary gland.

Replacing corn with FFFCG also increased the proportion of another important CLA isomer, *trans*-10, *cis*-12 CLA, in milk fat (Table 4). Higher concentrations of *trans*-10 C18:1 and *trans*-10, *cis*-12 CLA are commonly observed in milk from cows with MFD [3]. This condition is generally associated with a switch from the normal rumen biohydrogenation pathway to an increased synthesis of *trans*-10 C18:1 at the expense of *trans*-11 C18:1 [45], resulting in *trans*-10 C18:1/*trans*-11 C18:1 ratios above 1.0 (*trans*-10 shift) [46]. However, the magnitude of changes in the proportions of *trans*-10 C18:1 and *trans*-10, *cis*-12 CLA in milk fat from FFCG-fed cows does not appear to have been sufficient to induce MFD, as the milk fat content from cows supplemented with FFCG increased linearly (Table 2). As previously discussed, the absence of MFD appears to be associated with the composition of the basal diet used in our study.

The linear reduction in proportions of linolenic acid (C18:3 n-3), eicosapentaenoic acid (EPA, 20:5 n-3), and docosapentaenoic acid (DPA, 22:5 n-3) in milk fat (Table 3) can be explained by the extensive biohydrogenation of C18:3 n-3 in the rumen [47] and the negative effect of FFCG on DM intake (data not shown), which are consistent with the very low C18:3 n-3 content found in milk fat across dietary treatments (0.08 to 0.11 g/100 g total GA). In addition, the feedstuffs used in the experimental diets (especially sugarcane bagasse) have limited potential for enriching milk with C18:3 n-3 [3,48], unlike other tropical forages [49]. Milk C18:3 n-3 values as low as those found in the present study have been reported for crossbred cows consuming 30 to 54% sugarcane bagasse in their diet (on average 0.10 g/100 g FA) [48]. For cows fed fresh sugarcane, cactus cladodes, and FFCG, [3] reported milk C18:3 n-3 values between 0.20 and 0.18 g/100 g FA.

The effects of FFCG diets on milk C18:2 n-6 and C18:3 n-3 contents increased the n-6:n-3 ratio in milk significantly (Table 5). The n-6/n-3 ratio is a parameter used to evaluate the nutritional quality of fats, oils, and diets; foods with lower n-6/n-3 values are more desirable from a human-health perspective [50]. However, the lower proportions of medium-chain SFA and the enrichment of milk fat with MUFA and PUFA by FFCG diets, particularly with bioactive compounds such as *trans*-11 C18:1 and *cis*-9, *trans*-11 CLA, are considered benefits of using FFCG as an alternative replacement to corn in dairy cow diets.

## 5. Conclusions

Full-fat corn germ can be recommended as a substitute to ground corn for crossbred cows fed cactus cladodes and sugarcane bagasse and producing 15 kg of milk/day. Besides maximizing milk yield, FFCG reduces the saturated fatty acid content of milk and increases the proportions of desirable unsaturated fatty acids, such as *trans-11* C18:1 and *cis-9, trans-11* CLA. Therefore, the substitution of corn for FFCG may be an interesting strategy for the energy supplementation of herds located in the semi-arid region.

## Figures and Tables

**Table 1 animals-13-00568-t001:** Ingredient proportions and composition of experimental diets (g/kg DM).

Ingredient	Substitution Level of Corn for FFCG ^1^				
	0	25	50	75	100
Cactus cladodes	415.0	415.0	415.0	415.0	415.0
Sugarcane bagasse	318.8	319.1	319.2	319.4	319.5
Ground corn	148.0	111.0	74.0	37.0	0.0
FFCG ^1^	0.0	37.0	74.0	111.0	148.0
Corn gluten meal	83.0	83.0	83.0	83.0	83.0
Urea/Ammonium sulfate ^2^	18.2	17.9	17.8	17.6	17.5
Salt	5.0	5.0	5.0	5.0	5.0
Mineral blend ^3^	12.0	12.0	12.0	12.0	12.0
**Composition**					
Dry matter ^4^	431.4	434.3	437.1	439.8	442.7
Organic matter	924.7	924.8	924.9	924.9	924.9
Ether extract	16.6	30.9	45.2	59.5	73.8
Crude protein	140.2	141.2	142.5	143.5	144.9
cNDF ^5^	380.4	385.7	390.8	396.0	401.1
Non-fiber carbohydrates ^6^	467.5	446.1	425.0	403.7	382.6
Total digestible nutrients	686.8	715.3	718.3	744.3	753.4
Total fatty acids	10.7	23.6	36.8	49.9	62.9
C16:0	2.17	3.89	5.62	7.34	9.06
C18:0	0.42	0.70	0.98	1.27	1.55
*cis*-9 C18:1	2.52	7.09	11.66	16.23	20.79
C18:2 n-6	4.83	11.12	17.41	23.69	29.98
C18:3 n-3	1.15	1.27	1.38	1.50	1.61

^1^ Full-fat corn germ. ^2^ Urea + ammonium sulfate at 9:1 ratio. ^3^ Components: bicalcium phosphate, limestone, common salt, sulfur flower, zinc sulfate, copper sulfate, manganese sulfate, potassium iodate, and sodium selenite. ^4^ g/kg of fresh matter. ^5^ Neutral detergent fiber corrected for ash and protein. ^6^ Calculated according to the equation proposed by Reference [12].

**Table 2 animals-13-00568-t002:** Milk production and milk composition of cows fed diets containing increasing levels of FFCG in substitution to ground corn.

Item	Substitution Level of Corn for FFCG, %					SEM ^1^	*p*-Value ^2^	
	0	25	50	75	100		L	Q
Milk production, kg/day	13.37	13.97	14.61	14.93	14.08	1.123	0.0361	0.0183
Milk composition, %								
Fat	3.45	3.42	3.73	3.68	3.66	0.174	0.0387	0.4025
Protein	3.06	2.96	2.97	2.88	2.97	0.075	0.0062	0.0106
Total solids	11.80	11.59	12.03	11.91	11.88	0.206	0.1800	0.6387

^1^ Standard error of the mean. ^2^ L: linear effect; Q: quadratic effect. Linear and quadratic effects were considered statistically significant when *p* < 0.05.

**Table 3 animals-13-00568-t003:** Milk fatty acid (FA) profile (g/100 g total FA) of cows fed diets containing increasing levels of FFCG in substitution to ground corn.

Item	Substitution Level of Corn for FFCG, %					SEM ^1^	*p*-Value ^2^	
	0	25	50	75	100		L	Q
C4:0	3.3087	3.4743	3.4119	3.2028	3.1524	0.1602	0.1198	0.1898
C5:0	0.0204	0.0201	0.0185	0.0183	0.0188	0.0010	0.0803	0.3653
C6:0	2.2513	2.1748	1.9073	1.7353	1.5770	0.1276	<0.0001	0.7427
C7:0	0.0168	0.0140	0.0123	0.0113	0.0105	0.0010	<0.0001	0.1172
C8:0	1.4713	1.3555	1.1163	0.9766	0.8857	0.0875	<0.0001	0.3339
C9:0	0.0204	0.0156	0.0126	0.0127	0.0117	0.0012	<0.0001	0.0076
C10:0	3.1593	2.6774	2.0533	1.8146	1.5848	0.2063	<0.0001	0.0515
C11:0	0.0704	0.0546	0.0408	0.0369	0.0321	0.0059	<0.0001	0.0136
C12:0	3.6389	2.8887	2.1906	1.9675	1.6890	0.2204	<0.0001	0.0066
*cis-9* C12:1	0.1009	0.0794	0.0507	0.0480	0.0364	0.0064	<0.0001	0.0320
C14:0	11.0734	9.5893	7.9794	7.3498	6.5737	0.5637	<0.0001	0.1069
*iso* C14:0	0.1160	0.1023	0.0890	0.0913	0.0745	0.0061	<0.0001	0.5758
*cis-9* C14:1	1.0400	0.8188	0.6518	0.5534	0.4878	0.0739	<0.0001	0.0240
C15:0	1.0387	0.8699	0.7536	0.7121	0.6372	0.0304	<0.0001	0.0112
*iso* C15:0	0.2718	0.2318	0.2023	0.1832	0.1520	0.0120	<0.0001	0.4209
*anteiso* C15:0	0.5919	0.5183	0.4831	0.4400	0.3804	0.0236	<0.0001	0.7079
C16:0	31.5946	26.2426	23.1183	23.0576	21.4286	1.1107	<0.0001	0.0007
*iso* C16:0	0.2248	0.1887	0.1559	0.1496	0.1219	0.0105	<0.0001	0.1858
*trans-9* C16:1	0.0889	0.1520	0.2190	0.2691	0.3371	0.0291	<0.0001	0.9202
*cis-9* C16:1	1.5601	1.1298	0.9953	0.9084	0.8847	0.1175	<0.0001	0.0147
C17:0	0.6489	0.5407	0.4830	0.4644	0.4005	0.0160	<0.0001	0.0109
*iso* C17:0	0.4066	0.3674	0.3636	0.3451	0.3098	0.0171	<0.0001	0.8657
*anteiso* C17:0	0.5592	0.4903	0.4840	0.4610	0.3442	0.0446	0.0016	0.4852
*cis-9* C17:1	0.2496	0.1834	0.1719	0.1569	0.1329	0.0183	<0.0001	0.0091
C18:0	6.8171	10.6160	11.7213	13.3583	13.1279	0.8195	<0.0001	0.0027
C20:0	0.1308	0.1681	0.1753	0.1908	0.1836	0.0124	<0.0001	0.0019
C20:1	0.1263	0.1414	0.1460	0.1383	0.1461	0.0086	0.0157	0.1204
C20:2	0.0284	0.0268	0.0236	0.0224	0.0198	0.0015	<0.0001	1.0000
C20:3	0.0486	0.0492	0.0476	0.0477	0.0437	0.0031	0.0443	0.2475
C20:4	0.1947	0.1540	0.1273	0.1312	0.1048	0.0101	<0.0001	0.0284
C20:5	0.0140	0.0136	0.0122	0.0121	0.0120	0.0007	0.0004	0.8858
C21:0	0.0203	0.0200	0.0168	0.0178	0.0157	0.0014	0.0004	0.8604
C22:0	0.1207	0.1096	0.0927	0.0810	0.0799	0.0100	<0.0001	0.1267
C22:5	0.0331	0.0279	0.0246	0.0244	0.0201	0.0019	<0.0001	0.2958
C23:0	0.0234	0.0227	0.0185	0.0206	0.0193	0.0015	0.0257	0.3348
C24:0	0.0460	0.0399	0.0313	0.0312	0.0286	0.0033	<0.0001	0.0933

^1^ Standard error of the mean. ^2^ L: linear effect; Q: quadratic effect. Linear and quadratic effects were considered statistically significant when *p* < 0.05.

**Table 4 animals-13-00568-t004:** The proportion of C18 fatty acids (g/100 g total FA) in milk fat of cows fed diets containing increasing levels of FFCG in substitution to ground corn.

Item	Substitution Level of Corn for FFCG, %					SEM ^1^	*p*-Value ^2^	
	0	25	50	75	100		L	Q
C18:0	6.8171	10.6160	11.7213	13.3583	13.1279	0.8195	<0.0001	0.0027
*trans-4* C18:1	0.0276	0.0448	0.0805	0.0883	0.1019	0.0091	<0.0001	0.1928
*trans-5* C18:1	0.0228	0.0354	0.0614	0.0654	0.0779	0.0080	<0.0001	0.2399
*trans-6/trans-8* C18:1	0.2269	0.4231	0.6045	0.6456	0.8283	0.0570	<0.0001	0.2384
*trans-9* C18:1	0.1734	0.3130	0.4415	0.5258	0.6511	0.0517	<0.0001	0.6210
*trans-10* C18:1	0.3365	0.7232	1.1874	1.3262	1.7667	0.1900	<0.0001	0.7287
*trans-11* C18:1	1.3372	2.3243	3.5807	4.2577	5.3995	0.5546	<0.0001	0.8273
*trans-12* C18:1	0.2138	0.4682	0.6863	0.7362	0.9160	0.0567	<0.0001	0.0310
*trans-13/trans-14* C18:1	0.5020	1.0286	1.1966	1.2566	1.3210	0.1690	0.0012	0.1103
*trans-16* C18:1	0.1322	0.2573	0.3397	0.3700	0.3935	0.0319	<0.0001	0.0041
*cis-9* C18:1	17.0081	19.9223	22.5621	21.9779	22.9734	1.1373	<0.0001	0.0079
*cis-11* C18:1	0.7104	0.6519	0.7060	0.6928	0.6793	0.0472	0.7370	0.7622
*cis-12* C18:1	0.1968	0.3647	0.4960	0.4331	0.4591	0.0394	<0.0001	0.0004
*cis-13* C18:1	0.0891	0.0869	0.1094	0.1011	0.1195	0.0107	0.0110	0.7519
*cis-15* + 19:0 C18:1	0.0494	0.0600	0.0613	0.0629	0.0709	0.0061	0.0109	0.8088
*trans-9, trans-12* C18:2	0.0213	0.0181	0.0246	0.0369	0.0297	0.0043	0.0041	0.8717
*cis-9, trans-12* C18:2	0.0315	0.0530	0.0505	0.0487	0.0760	0.0080	<0.0001	0.5459
C18:2 n-6	2.8782	2.8602	2.9992	3.0359	3.0295	0.1223	0.0350	0.7591
*cis-9, trans-11* C18:2	0.7576	1.2476	1.9179	1.9106	2.6108	0.2649	<0.0001	0.7088
*trans-9, cis-11* C18:2	0.0200	0.0266	0.0351	0.0337	0.0403	0.0059	0.0028	0.5700
*trans-10, cis-12* C18:2	0.0142	0.0192	0.0190	0.0197	0.0218	0.0021	0.0060	0.4464
C18:3 n-6	0.0492	0.0486	0.0419	0.0454	0.0442	0.0042	0.0900	0.3202
C18:3 n-3	0.1104	0.0952	0.0957	0.0877	0.0780	0.0073	0.0001	0.9021

^1^ Standard error of the mean. ^2^ L: linear effect; Q: quadratic effect. Linear and quadratic effects were considered statistically significant when *p* < 0.05.

**Table 5 animals-13-00568-t005:** Proportions of main fatty acid (FA) groups (g/100 g of total FA), FA ratios, and stearoyl-CoA desaturase 1 (SCD-1) indices in milk fat of cows fed diets containing increasing levels of FFCG in substitution to ground corn.

Item	Substitution Level of Corn for FFCG, %					SEM ^1^	*p*-Value ^2^	
	0	25	50	75	100		L	Q
∑ n-3 FA	0.1577	0.1368	0.1325	0.1243	0.1096	0.0078	<0.0001	0.7042
∑ n-6 FA	3.1992	3.1387	2.2394	3.2822	3.2421	0.1265	0.3048	0.9479
∑ *trans* C18:1	2.9723	5.6185	8.1786	9.2714	11.4560	0.8666	<0.0001	0.2944
∑ *trans* C18:1—(VA + RA) ^3^	0.8774	2.0466	2.6798	3.1034	3.4456	0.3297	<0.0001	0.1104
∑ SCFA ^3^	10.1904	9.6823	8.4889	7.7291	7.2003	0.5379	<0.0001	0.6466
∑ MCFA ^3^	46.3070	38.7203	33.2884	32.3748	29.6913	1.7512	<0.0001	0.0019
∑ SFA ^3^	67.7875	62.9044	57.0342	56.8236	52.9263	1.8028	<0.0001	0.0717
∑ MUFA ^3^	24.6638	29.5881	34.6628	34.9050	38.0316	1.5307	<0.0001	0.0229
∑ *cis* MUFA	21.3750	23.6150	26.0620	25.1550	26.0400	1.1212	<0.0001	0.0160
∑ *trans* MUFA	3.2900	5.9720	8.5980	9.7500	11.9930	0.8927	<0.0001	0.3146
∑ PUFA ^3^	4.2270	4.6760	5.4560	5.4950	6.1690	0.3233	<0.0001	0.6946
∑ OCFA ^3^	1.9463	1.6186	1.4122	1.3447	1.1938	0.0488	<0.0001	0.0034
∑ BCFA ^3^	2.2287	1.9492	1.8241	1.7135	1.4212	0.0744	<0.0001	0.9606
∑ OBCFA ^4^	4.1752	3.5681	3.2359	3.0584	2.6147	0.1142	<0.0001	0.1662
**FA Ratios**								
*trans* C18:1/C18:0	0.3848	0.5326	0.7070	0.6794	0.8989	0.0768	<0.0001	0.7984
*trans-11* C18:1/C18:0	0.1729	0.2180	0.3064	0.3134	0.4208	0.0448	<0.0001	0.6996
n-6/n-3	20.5366	23.3382	24.8491	27.1740	29.7705	1.1662	<0.0001	0.9052
**SCD-1 Indices ^5^**								
SCD14	0.0858	0.0788	0.0761	0.0679	0.0688	0.0052	<0.0001	0.3058
SCD16	0.0457	0.0408	0.0414	0.0370	0.0392	0.0031	0.0177	0.2531
SCD18	0.7288	0.6520	0.6553	0.6247	0.6373	0.0199	0.0004	0.0254
SCDCLA	0.3710	0.3553	0.3469	0.3194	0.3281	0.0157	0.0024	0.4959

^1^ Standard error of the mean. ^2^ L: linear effect; Q: quadratic effect. Linear and quadratic effects were considered statistically significant when *p* < 0.05. ^3^ VA = *trans*-11 C18:1 (vaccenic acid); RA = *cis*-9, *trans*-11 CLA (rumenic acid); SCFA = short-chain fatty acids; MCFA = medium-chain fatty acids; SFA = saturated fatty acids; MUFA = monounsaturated fatty acids; PUFA = polyunsaturated fatty acids; OCFA = odd-chain fatty acids. ^4^ Sum of odd- and branched-chain fatty acids, except for 13:0, iso 17:0, and *anteiso* 17:0, as these co-eluted with *cis*-9 12:1, *trans*-9 16:1, and *cis*-9 16:1, respectively. ^5^ Stearoyl-CoA desaturase-1 (SCD-1) indices calculated for cis-9 14:1/14:0 (SCD_14_), *cis*-9 16:1/16:0 (SCD_16_), cis-9 18:1/18:0 (SCD_18_), and *cis*-9, *trans*-11 CLA/*trans*-11 18:1 (SCD_CLA_) [17].

## Data Availability

Not applicable.

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
