# Peer review of "Nutritional Quality of Milk Fat from Cows Fed Full-Fat Corn Germ in Diets Containing Cactus Opuntia and Sugarcane Bagasse as Forage Sources"

_animals, 2023, doi:10.3390/ani13040568_

Round 1

Reviewer 1 Report

Title: The title is not concise and requires to be rewritten to be clearer (too long).

Simple Summary:  Lines 18 and 19, ‘However, feeding traditional oils, such as soybean or sunflower oil, may be disadvantageous’ This sentence is out of place, should be removed, and does not add value to the summary.

Lines 27-28, ‘Hence, dairy producers and butter manufacturers can increase profit by including FFCG in the diet while benefiting consumers with a CLA-enriched milk’. This sentence needs to be rewritten. Dairy producers and butter manufacturers perform different duties.

Abstract: The author needs to state the hypothesis being tested in the abstract.

Lines 38-41, ‘This strategy would add value to milk products in a scenario of commercial valorization of the nutritional quality of milk fat. In addition, the milk fat response indicates that the basal diet was favorable to the rumen environment, preventing the trans-10 shift commonly associated with milk fat depression. What do you mean by, ‘This strategy would add value to milk products in a scenario of commercial valorization’? Please rewrite the sentence to make it clearer.

Introduction:

Good

Please rewrite line 74, ‘cactus cladodes (Opuntia stricta 74 [Haw.] Haw)’.

Materials and Methods:

The author has indicated that the experiment took place from January to May 2020. However, lines 86-88 do not add up or make sense, ‘Subsequently, the cows were distributed into two 5 x 5 Latin Squares with 21-day experimental periods; within a period, the first 14 days were used for adaptation of the cows to experimental diets and the last 7 days for data and sample collection’. Please clarify.

Why did you start the experiment when the cows were 90 ± 15 days in milk?

What informed your decision about the different dietary treatment levels of substitution at 0%, 25%, 50%, 96 75%, and 100% of ground corn for full-fat corn germ (FFCG)? Because at 100% substitution, the ether extract is more than 7% required for a dairy cow.

Lines 97-98, ‘Cactus cladodes (Opuntia stricta [Haw.] Haw)’.

Table 1, replace Ureia with Urea.

Results

Good

Table 2, you can write the abbreviated FFCG instead of the full name

Where are the results for casein, lactose, total solids, and urea as indicated in lines 123-124?

Where is the result for the fatty acid profile of the diet fed to the experimental cows?

Discussion:

Thorough and coherent discussion

Conclusions

Good conclusion

Reference

Good list of references.

Author Response

TITLE: The title is not concise and requires to be rewritten to be clearer (too long).

The title has been rewritten as “Nutritional quality of milk fat from cows fed full-fat corn germ in diets containing cactus Opuntia and sugarcane bagasse as forage sources”.

SIMPLE SUMMARY:

Lines 18 and 19, ‘However, feeding traditional oils, such as soybean or sunflower oil, may be disadvantageous’ This sentence is out of place, should be removed, and does not add value to the summary.

Response: Sentence removed.

Lines 27-28, ‘Hence, dairy producers and butter manufacturers can increase profit by including FFCG in the diet while benefiting consumers with a CLA-enriched milk’. This sentence needs to be rewritten. Dairy producers and butter manufacturers perform different duties.

Response: rewritten as “Hence, FFCG supplementation may be advantageous to milk producers through improved cow performance while promoting greater fat yield and fat quality for manufacturing of dairy products with potential health benefits. (Lines 26-28)

ABSTRACT:

The author needs to state the hypothesis being tested in the abstract.

Response: Revised as suggested. Lines 30-31: “…We hypothesized that ground corn can be effectively replaced by FFCG when cactus cladodes and sugarcane bagasse are used as forage sources.”

Lines 38-41, ‘This strategy would add value to milk products in a scenario of commercial valorization of the nutritional quality of milk fat. In addition, the milk fat response indicates that the basal diet was favorable to the rumen environment, preventing the trans-10 shift commonly associated with milk fat depression. What do you mean by, ‘This strategy would add value to milk products in a scenario of commercial valorization’? Please rewrite the sentence to make it clearer.

Response: Revised. Lines 37-41: “We conclude that the substitution of corn for FFCG in diets based on cactus cladodes and sugarcane bagasse positively modifies the FA profile of milk and could add commercial value to milk products (e.g., CLA-enriched milk).”

INTRODUCTION:

Please rewrite line 74, ‘cactus cladodes (Opuntia stricta 74 [Haw.] Haw)’.

Response: the scientific name Opuntia stricta (Haw.) Haw was written according to previous literature (Hosking et al., 1994) and plant databases.

Hosking, J.R., Sullivan, P.R., Welsb, S.M. Biological control of Opuntia stricta (Haw.) Haw. var. stricta using Dactylopius opuntiae (Cockerell) in an area of New South Wales, Australia, where Cactoblastis cactorum (Berg) is not a successful biological control agent. Agriculture, Ecosystems & Environment, 48:241-255, 1994. doi: https://doi.org/10.1016/0167-8809(94)90106-6
Royal Botanic Gardens: Plants of the Word Online: https://powo.science.kew.org/taxon/urn:lsid:ipni.org:names:137078-1#publications.

MATERIALS AND METHODS:

The author has indicated that the experiment took place from January to May 2020. However, lines 86-88 do not add up or make sense, ‘Subsequently, the cows were distributed into two 5 x 5 Latin Squares with 21-day experimental periods; within a period, the first 14 days were used for adaptation of the cows to experimental diets and the last 7 days for data and sample collection’. Please clarify.

Response: A 5x5 Latin square design is the arrangement of 5 treatments (in our case diets), each one repeated 5 times (periods). Each of our experimental periods had 21 days (14 days used for adaptation followed by 7 days of data collection). Therefore, the whole experiment was carried out for 105 days, starting on January 27th and ending on May 10th. The text was revised accordingly: Lines 93 (“…from January 27 to May 10, 2020.”) and lines 101-102 (“…Thus, the total experimental period lasted 105 days.”).

Why did you start the experiment when the cows were 90 ± 15 days in milk?

Response: We intended to avoid the metabolic effects of early lactation on performance, milk composition, and milk fatty acid profile because these effects of early lactation were not our interest at this time. In this case, the current recommendation is to start the trial when cows had passed the lactation peak and finish by mid-pregnancy.

What informed your decision about the different dietary treatment levels of substitution at 0%, 25%, 50%, 96 75%, and 100% of ground corn for full-fat corn germ (FFCG)? Because at 100% substitution, the ether extract is more than 7% required for a dairy cow.

Response: We appreciate your observation. However, because there were no reports providing recommendations for inclusion of full-fat corn germ in dairy cow diets in tropical conditions and we did not know the availability of full-fat corn germ to rumen biohydrogenation, we needed to utilize high levels of full-fat corn germ to find its threshold and provide recommendations for an adequate and practical inclusion level. The levels described in this manuscript were useful for planning follow-up experiments.

Lines 97-98, ‘Cactus cladodes (Opuntia stricta [Haw.] Haw)’.

Response: Name of the cactus species deleted (Line 110).

Table 1, replace Ureia with Urea.

Response: Replaced as requested.

RESULTS:

Table 2, you can write the abbreviated FFCG instead of the full name.

Response: Full name deleted from tables 2 – 5 and added to Line 172 for being the first mention to FFCG in the Results section.

Where are the results for casein, lactose, total solids, and urea as indicated in lines 123-124?

Response: Mentions of casein, lactose, and urea have been removed from the Material and Methods section (line 142).

Where is the result for the fatty acid profile of the diet fed to the experimental cows?

Response: the FA acid profile of the diets has been added to Table 1 as suggested.

Reviewer 2 Report

Manuscipt title: Full-fat corn germ as a lipid supplement for dairy cows fed  Opuntia cactus cladodes and sugarcane bagasse as forage  sources: milk production and nutritional quality of milk fat.

Manuscript is interesting and something news about changes in diet in Brasilian low producting cows according with nutrition condution in this part of Brasil.

Results showed that changes in diet increase milk production and develop a milk fat quality  (better milk fatty acids composition) than in control group.

Manuscripi is very good written in the all part of text. I have not any significant comment.

Author Response

We highly appreciate your evaluation.